# SAW Resonators and Filters Based on Sc_0.43_Al_0.57_N on Single Crystal and Polycrystalline Diamond

**DOI:** 10.3390/mi13071061

**Published:** 2022-06-30

**Authors:** Miguel Sinusia Lozano, Laura Fernández-García, David López-Romero, Oliver A. Williams, Gonzalo F. Iriarte

**Affiliations:** 1Institute for Optoelectronic Systems and Microtechnology, Universidad Politécnica de Madrid, Avenida Complutense, 30, 28040 Madrid, Spain; 2Nanophotonics Technology Center, Universitat Politècnica de València, Camino de Vera, s/n Edificio 8F|Planta 2ª, 46022 Valencia, Spain; msinloz@ntc.upv.es; 3Departamento Ciencia de Materiales, Escuela Técnica Superior de Ingenieros de Caminos, Canales y Puertos, Universidad Politécnica de Madrid, Ciudad Universitaria, Calle del Profesor Aranguren 3, 28040 Madrid, Spain; 4Departamento de Sensores y Sistemas de Ultrasonidos, Instituto de Tecnologías Físicas y de la Información Leonardo Torres Quevedo—ITEFI, CSIC, Calle Serrano 144, 28006 Madrid, Spain; laura.fernandez@csic.es; 5Instituto de Micro y Nanotecnología, IMN-CNM, CSIC Isaac Newton, 8, Tres Cantos, 28760 Madrid, Spain; david.lopezromero@csic.es; 6School of Physics and Astronomy, Cardiff University, Cardiff CF24 3AA, UK; williamso@cardiff.ac.uk

**Keywords:** SAW devices, piezoelectricity, ScAlN thin film, diamond thin film, 5G technology, electromechanical coupling coefficient k2, Q-factor

## Abstract

The massive data transfer rates of nowadays mobile communication technologies demand devices not only with outstanding electric performances but with example stability in a wide range of conditions. Surface acoustic wave (SAW) devices provide a high Q-factor and properties inherent to the employed materials: thermal and chemical stability or low propagation losses. SAW resonators and filters based on ${\mathrm{Sc}}_{0.43}{\mathrm{Al}}_{0.57}N$ synthetized by reactive magnetron sputtering on single crystal and polycrystalline diamond substrates were fabricated and evaluated. Our SAW resonators showed high electromechanical coupling coefficients for Rayleigh and Sezawa modes, propagating at 1.2 GHz and 2.3 GHz, respectively. Finally, SAW filters were fabricated on ${\mathrm{Sc}}_{0.43}{\mathrm{Al}}_{0.57}N$/diamond heterostructures, with working frequencies above 4.7 GHz and ~200 MHz bandwidths, confirming that these devices are promising candidates in developing 5G technology.

## 1. Introduction

Surface acoustic wave (SAW) devices leverage the piezoelectric effect to generate and detect electroacoustic signals [1,2]. These devices are fabricated with cutting-edge clean room technologies, which significantly shrinks their unitary costs. The possibility of controlling phonons and monitoring their interaction with electrons, photons, or magnetic spins have attracted the attention of the research community. Their applications have increased from the delay lines of the early 1960s to the nowadays state-of-the-art sensors or the emerging quantum technologies [3,4]. Within the scope of mobile telecommunication, the operating frequencies of these devices have been steadily increasing from 450 MHz to 6 GHz. Furthermore, SAW devices based on polycrystalline piezoelectric thin films are a cost-effective MEMS solution for the 5G technological constraints due to their outstanding capabilities such as power handling or thermal stability [5,6].

Among polycrystalline piezoelectric thin films, AlN has been extensively studied for its high SAW propagation velocity due to the stiffness of the compound, as well as its thermal and chemical stability [7,8]. However, its relatively low piezoelectric constant d_33_ restrains its applicability where large electromechanical coupling coefficients ($k^{2}$) are required. In this regard, the introduction of Sc atoms into the wurtzite AlN structure increases the piezoelectric response of the thin film [9]. The maximum increase is reported to occur at an Sc concentration of 43%. However, experimentally, the synthesis of this compound has been challenging for the competitive synthesis of the rock−salt phase (non-piezoelectric) of ScAlN, which is more energetically favorable at Sc concentrations above 55% [10,11,12]. Furthermore, the inclusion of Sc does not only increase the piezoelectric response of the compound but reduces its elastic constant and alters the optoelectronic properties, such as the bandgap [13,14].

In the case of loss-less materials, the electromechanical coupling coefficient ($k^{2}$) is a measure of the conversion efficiency between the mechanical and electrical energies and vice versa, and it is directly related to the piezoelectric response of the thin film [15]. Therefore, the ScAlN compound and its increase of the piezoelectric constant with the Sc concentration is a promising material to fabricate high-frequency SAW devices with outstanding performances.

In this work, SAW devices were fabricated with ${\mathrm{Sc}}_{0.43}{\mathrm{Al}}_{0.57}N$/diamond-based heterostructures. Highly c-axis oriented ${\mathrm{Sc}}_{0.43}{\mathrm{Al}}_{0.57}N$ thin films were synthesized on polycrystalline and single crystal diamond substrates. The electroacoustic properties of these layered structures were assessed. From the 1-port resonators, the effective electromechanical coupling coefficients ($K_{\mathrm{eff}}^{2})$ and effective propagation velocities were extracted. Finally, SAW filters working at frequencies above 4.70 GHz are presented.

## 2. Materials and Methods

The synthesis of the piezoelectric thin film was carried out using a home-built reactive magnetron sputtering system. The target alloy (${\mathrm{Sc}}_{0.6}{\mathrm{Al}}_{0.4}$) was placed 45 mm from the substrate. The process was carried out without intentional heating, and the temperature was monitored using a K-type thermocouple located below the substrate holder. The magnetron was powered using a pulsed DC generator (ENI RPG50), with a pulse width set to 1616 ns.

The polycrystalline diamond (PCD) substrates were synthesized by microwave plasma chemical vapor deposition (MPCVD) on a 500-µm thick Si (001) supporting layer. The single crystal diamond (111) (SCD) was purchased (EDP Corporation, Osaka, Japan) for evaluating its influence on the SAW propagation characteristics.

Prior to the ScAlN synthesis process, the substrates were cleaned using the following two-solvent method. First, the substrates were rinsed in acetone at 60 °C, followed by a sonication bath in methanol, both for 5 min. Afterwards, the samples were blown dry with nitrogen ($N_{2}$) and introduced into the load-lock chamber.

The synthesis chamber was conditioned for three minutes in a pure Argon (Ar) atmosphere (30 sccm), with a discharge power of 500 W and a process pressure of 1.33 Pa. Nitrogen was then introduced, the gas admixture ratio ($N_{2}$/($N_{2}$ + Ar)) was adjusted to 25%, and the process pressure and discharge power were set to 1.33 Pa and 500 W, respectively, for 3 min. Finally, the process pressure was adjusted to 0.40 Pa. After 3 min, the plasma was turned off, and the substrates were transferred from the load-lock chamber. Afterwards, with the shutter closed, the plasma was ignited again using the synthesis conditions (0.40 Pa, 500 W, 25% gas admixture ratio), and after two minutes, the shutter was moved away, beginning the synthesis process.

The degree of c-axis orientation of the synthesized piezoelectric thin film was assessed via 0002 ω-θ scans using X-ray diffraction (XRD, Phillips X-Pert Pro MRD diffractometer), with Cu ${K\alpha }_{1}$ radiation (λ = 1.54059 Å), 45 kV, and 40 mA.

The interdigital transducers (IDT) were fabricated using a standard lift-off process. They were patterned with an e-beam lithography system (Crestec CABL-9500C, Hachioji, Japan). Due to the insulating behavior of the ScAlN thin film and the underlying polycrystalline diamond substrates, an organic anti-static layer (Espacer 300Z, Showa Denko K.K, Tokyo, Japan) was spun on top of the e-beam resist to avoid charge accumulation.

The fabricated 1-port resonators comprise a 2000-nm ${\mathrm{Sc}}_{0.43}{\mathrm{Al}}_{0.57}N$ thin film and 110-nm thick (700 nm width) Au IDT (λ = 2.8 µm). The filters were fabricated using an 850-nm thick ${\mathrm{Sc}}_{0.43}{\mathrm{Al}}_{0.57}N$ thin film and 300-nm wide Au IDT (λ = 1.2 µm). In this case, the Au thickness of the filters IDT were different: 130 nm and 65 nm thick for the PCD and SCD, respectively. A 5-nm thick Cr layer was employed as an adhesion layer. In all devices, a 0.5 metallization ratio was employed (Figure 1). An SEM inspection of the fabricated SAW filters can be found in (Appendix A) The 1-port resonators had 60 finger pairs and two grounded reflectors with 60 finger pairs with the designed IDT wavelength. On the other hand, the π-type SAW filters comprised 3 1-port resonators.

The electrical characterization of the devices was carried out using a vector network analyzer (Agilent N5230 A, Santa Clara, CA, USA), with standard, 300-µm pitch, ground-source-ground (GSG) probes (Picoprobe 40A; C style adaptor, GGB Industries, Inc, Naples, FL, USA). A standard short, open, load, through (SOLT), 50 Ω, one-port calibration was employed. In the case of the 1-port resonators, the measurement resolution was set to 16,001 points in the 0.50 GHz to 6.00 GHz frequency range, whereas the measurement of the fabricated SAW filters was carried out with a resolution of 16,001 points in the 1.00 GHz to 10 GHz frequency range. In both cases, the output power was set to 0 dBm. Afterwards, the data processing and analysis was carried out using scikit-rf, an open-source Python package [16].

## 3. Results

The devices presented in this work targeted a ratio between piezoelectric thin film thickness and an IDT wavelength (d/λ) of 0.71, which is reported to provide the maximum $K_{eff}^{2}$ of the Sezawa mode for diamond-based heterostructures [17]. Additionally, in order to efficiently propagate the Sezawa mode at frequencies above 4 GHz, the designed IDT wavelength of the filters (1200 nm) was reduced compared to that of the resonators (2800 nm). In these devices, the resonance frequency was determined by the IDT wavelength and the propagation velocity of the wave through the layered structure [18]. Therefore, reducing the thickness of the piezoelectric thin film forces the wave to propagate predominantly through the diamond layer, leveraging its faster propagation velocity.

The single crystal diamond datasheet reported a surface roughness $R_{a}$ value below 2 nm and the AFM analyses carried out on the polycrystalline diamond substrate reported $R_{RMS}$ values below 2 nm. Regardless of the substrate, the full width at half maximum (FWHM) below 3° of the XRD 0002 ω-θ scans confirmed that the polycrystalline ScAlN thin films are highly c-axis oriented. The RBS analysis reported an Sc 43% concentration within the thin film (Appendix A).

### 3.1. Resonators

The reflection coefficient of the polycrystalline diamond resonators showed that several resonance frequency modes are generated together with second and third order reflections (Figure 2A). Among these, the reflection coefficient of the Rayleigh mode (1.20 GHz) outstands (~−40 dB). The Sezawa mode (2.06 GHz) and the second order Rayleigh mode (2.30 GHz) propagated with reflection coefficients below −15 dB.

The devices using the SCD substrate reported lower insertion losses (Figure 2B). Furthermore, in these devices, the reflection coefficient of the Sezawa mode (2.03 GHz) below −45 dB indicated the efficient excitation of this mode. The Rayleigh mode (1.21 GHz) propagated with a reflection coefficient below −5 dB, whereas its second order reflection mode (2.25 GHz) propagated with a reflection coefficient below −15 dB. In both heterostructures, higher reflection modes propagated above 2.50 GHz.

From the admittance characteristics (Figure 2C,D), the series ($f_{s}$) and parallel ($f_{p}$) frequencies can be extracted and then the effective propagation velocity ($v_{\mathrm{eff}}$) (Equation (1)) can be calculated, as can the effective electromechanical coupling coefficient ($K_{\mathrm{eff}}^{2}$) (Equation (2)), where λ is the designed IDT wavelength (Table 1) [19,20].
(1)$$v_{\mathrm{eff}}=\frac{f_{p}-f_{s}}{2}\lambda$$
(2)$$K_{\mathrm{eff}}^{2}=\frac{\pi^{2}}{8}\frac{f_{p}^{2}-f_{s}^{2}}{f_{s}^{2}}$$ A figure of merit (Table 2), usually employed to compare the performance of SAW devices, multiplied the Bode Q-factor (Equation (3)) by the electromechanical coupling coefficient. The Q-factor (Equation (4)) is defined as the ratio between the series or parallel resonance frequency and the −3-dB width of the frequency [21,22,23].
(3)$${\mathrm{FOM}}_{s,p}={\;K}_{\mathrm{eff}}^{2}\;*\;Q_{s,p}$$
(4)$$Q_{s,p}=\frac{f_{s,p}}{\Delta f_{-3\mathrm{dB}\left(s,p\right)}}$$

### 3.2. Filters

The transmission coefficient of the SAW filter fabricated using the PCD heterostructure showed that the Rayleigh and Sezawa modes propagated with their center frequency ($f_{c}$) at 2.66 GHz and 4.88 GHz, respectively (Figure 3A). Both modes presented a maximum gain above −6 dB, whereas their cut−off frequencies were below −12 dB. The −3-dB bandwidth of the Rayleigh mode was above 100 MHz, whereas the Sezawa mode bandwidth was above 180 MHz (Table 3).

On the other hand, the Rayleigh mode generated in the SAW filter fabricated using the SCD substrate had a band pass maximum of −2.5 dB (Figure 3B). The center frequency of this mode was 2.17 GHz. whereas the Sezawa mode $f_{c}$ was at 4.74 GHz, and its band pass maximum was −5 dB. The cut-off of both the lower ($f_{1}$) and higher ($f_{2}$) frequencies in the Rayleigh mode were below −18 dB, whereas those corresponding to the Sezawa mode were below −10 dB.

## 4. Discussion

In layered structures, the effective propagation velocity of a SAW (Equation (5)) depends on the elastic modulus (E) and density (ρ) of the particular thin films, which through these the wave propagates [24]. In this regard, both types of diamond substrates have similar stiffness, as the resonance frequency of the SAW modes in both resonators is comparable. Furthermore, when comparing the resonance frequencies with our previous works (${\mathrm{Sc}}_{0.26}{\mathrm{Al}}_{0.74}N$/PCD with Pt electrodes), the softening of the piezoelectric thin film with the Sc concentration was apparent, as the resonance frequency of both Rayleigh and Sezawa modes was reduced [25].
(5)$$v_{\mathrm{eff}}=\sqrt{\frac{E}{\rho }}$$

However, the single crystal diamond substrate shifted the conductance baseline of the device towards ~0 S (Figure 2D), which showed how the scattering of the SAW wave due to defects such as grain boundaries or voids deteriorated the electrical response of the devices. This is in agreement with previous works, where the grain size of the polycrystalline diamond substrates increased the insertion losses of the devices [26,27,28].

Focusing on the piezoelectric thin film, particularly in its defects and how they degrade the electrical response of the devices [29], most of the defects in the thin films accumulated within the proximity of layer interfaces, mainly due to lattice mismatches and strain relaxation mechanisms, which usually require heat treatments to be minimized. However, the propagation characteristics of our devices validated the characterization via XRD analysis and the high quality ${\mathrm{Sc}}_{0.43}{\mathrm{Al}}_{0.57}N$ thin films which, as previously mentioned, were synthesized with no intentional heating of the substrate, on both diamond substrates.

The electromechanical coupling coefficients lay between those reported previously in the literature (Table 4). However, there have been several approaches that can be undertaken to increase this value, such as adequate device design or the selection of the substrate or the electrode metal [17,30].

On the other hand, the structure with embedded electrodes (ScAlN/IDT/diamond) has been reported to considerably increase the $K_{\mathrm{eff}}^{2}$ coefficient [31]. Additionally, in order to boost the SAW propagation, a-plane (instead of c-plane) ScAlN thin films have been recently employed [32].

**Table 4 micromachines-13-01061-t004:** Comparison of resonance frequencies, compound composition, and electromechanical coupling coefficient.

ScAlN Composition	Synthesis Technique	Target	Substrate	Resonance Frequency (GHz)	Electrode Metal	$K_{eff}^{2}\;(\%)$	Q
${\mathrm{Sc}}_{0.11}{\mathrm{Al}}_{0.89}N$ [29]	MBE	-	Si	3.6	Ti/Au	3.7	146
${\mathrm{Sc}}_{0.27}{\mathrm{Al}}_{0.73}N$ [17]	RF	${\mathrm{Sc}}_{0.40}{\mathrm{Al}}_{0.60}$	PCD	2.5−3.5	Al/Cr	5.5–4.5	396–227
${\mathrm{Sc}}_{0.27}{\mathrm{Al}}_{0.73}N$ [30]	DC	ScAlN alloy	Si	0.2–0.3	Ti/Au	2	100
${\mathrm{Sc}}_{0.26}{\mathrm{Al}}_{0.74}N$ [33]	Pulsed DC	${\mathrm{Sc}}_{0.40}{\mathrm{Al}}_{0.60}$	Si	R 1.4	Pt	0.5	140
${\mathrm{Sc}}_{0.26}{\mathrm{Al}}_{0.74}N$ [18]	Pulsed DC	${\mathrm{Sc}}_{0.40}{\mathrm{Al}}_{0.60}$	PCD	R 1.5–S 2.6	Pt	2.8	R167–S180
${\mathrm{Sc}}_{0.43}{\mathrm{Al}}_{0.57}N$ [34]	RF	Dual	SCD SCD	3.75 2.9	Cu Cu	6.1 3.8	520 -
${\mathrm{Sc}}_{0.23}{\mathrm{Al}}_{0.77}N$ [32]	Pulsed DC	Dual (Al + Sc targets)	Sapphire	1.9–1.7	Pt	1.3–2.4	659–538
This work	Pulsed DC	${\mathrm{Sc}}_{0.60}{\mathrm{Al}}_{0.40}$	PCD&SCD	R 1.2–S 2.03	Cr/Au	3.2–3.7	R 250–S ~50

The filter bandwidth was, without some other circuit elements, constrained by the effective electromechanical coupling coefficient ($K_{\mathrm{eff}}^{2}$), as it ultimately constrained the separation between the series and parallel resonance frequencies.

The −3-dB bandwidth of the Sezawa mode in both the layered structures was above 180 MHz at the center frequencies of 4.70 GHz and 4.90 GHz for the PCD and SCD structures, respectively.

Higher bandwidth for both Rayleigh and Sezawa propagation modes were obtained in our ${\mathrm{Sc}}_{0.43}{\mathrm{Al}}_{0.57}N$-based heterostructures compared to AlN and ZnO based filters (Table 5). Furthermore, the insertion losses of our devices were similar to these polycrystalline piezoelectric thin films. However, the insertion losses of the devices based on single crystal piezoelectric thin films were lower, showing the importance of the device design and the quality of the polycrystalline thin film in the electrical response of the filters.

## 5. Conclusions

SAW devices have been fabricated using ${\mathrm{Sc}}_{0.43}{\mathrm{Al}}_{0.57}N$ as a piezoelectric thin film, synthetized over polycrystalline and single crystalline diamond substrates. The admittance characteristics confirm that ${\mathrm{Sc}}_{0.43}{\mathrm{Al}}_{0.57}N$ thin films can be employed in the fabrication of SAW devices with low insertion losses above 2 GHz. The higher insertion loss and attenuation of the SAW in the PCD structure revealed the damaging influence of the electrical performance of the scattering in the grain boundaries, as well as the defects within the crystal structure.

The ${\mathrm{Sc}}_{0.43}{\mathrm{Al}}_{0.57}N$ thin films employed in these devices were synthesized by reactive magnetron sputtering without the intentional heating of the substrate. The energy required to minimize the defects within the thin film is provided by the plasma conditions. The Rayleigh and Sezawa mode $K_{\mathrm{eff}}^{2}$ values substantially increased with the devices comprising the ${\mathrm{Sc}}_{0.43}{\mathrm{Al}}_{0.57}N$ thin films, as compared to those values obtained for the modes propagating within the ${\mathrm{Sc}}_{0.27}{\mathrm{Al}}_{0.73}N$ thin films shown in our previous works [18,39]. These two propagating modes are efficiently excited where the reflection coefficient of the Sezawa mode propagating in the SCD heterostructure outstands, with an attenuation close to −50 dB. Additionally, SAW filters with –3-dB bandwidth above 180 MHz have been fabricated at 4.7 GHz resonance frequencies with insertion losses below –5 dB with a SCD based device, revealing their potential application to 5G technology.

Furthermore, the devices presented in this work showed promising electrical performances for sensing applications where the AlN and ScAlN thermal, chemical, or high stiffness of the compound were exploited. Therefore, the applicability of these devices will not only be constrained by 5G technology, but these results reveal the potential of the versatile ScAlN compound for the next-generation of SAW devices.

## Figures and Tables

**Figure 1 micromachines-13-01061-f001:**
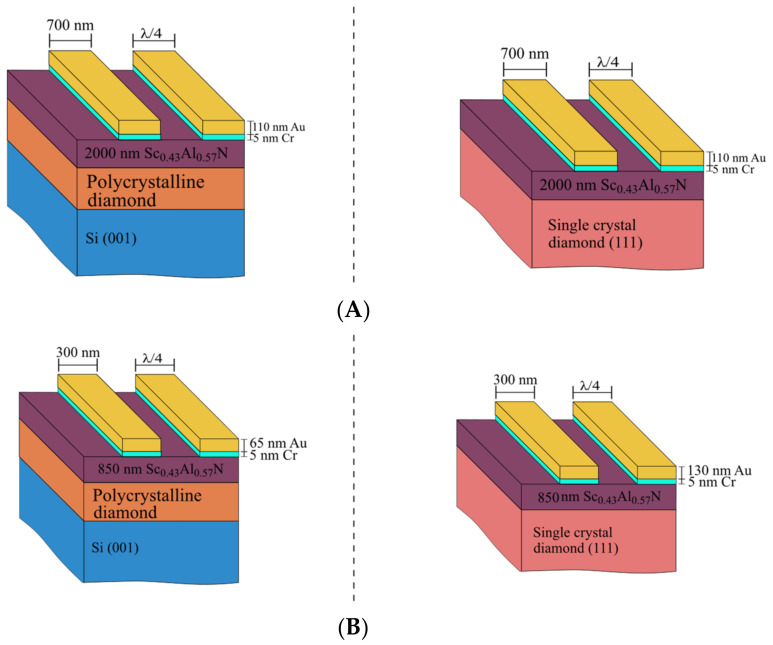
${\mathrm{Sc}}_{0.43}{\mathrm{Al}}_{0.57}N$ heterostructures of the fabricated devices with polycrystalline and single crystal diamond substrates. (**A**) The 1-port SAW resonators and (**B**) SAW filters.

**Figure 2 micromachines-13-01061-f002:**
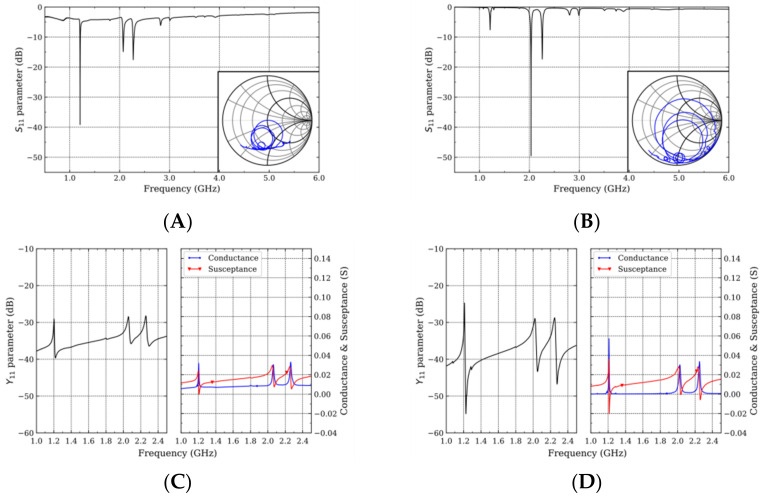
Measured reflection coefficient (inset: Smith chart) and admittance characteristics of the one-port resonators fabricated with the ${\mathrm{Sc}}_{0.43}{\mathrm{Al}}_{0.57}N$/polycrystalline (**A**,**C**) and single crystal diamond (**B**,**D**) heterostructures.

**Figure 3 micromachines-13-01061-f003:**
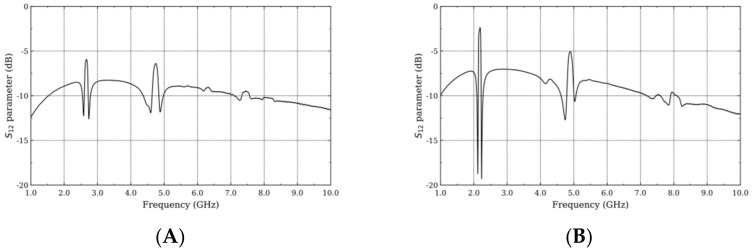
Transmission coefficient of a ladder-type SAW filter fabricated on ${\mathrm{Sc}}_{0.43}{\mathrm{Al}}_{0.57}N$/PCD (**A**) and ${\mathrm{Sc}}_{0.43}{\mathrm{Al}}_{0.57}N$/SCD (**B**) heterostructures.

**Table 1 micromachines-13-01061-t001:** Effective velocity ($v_{\mathrm{eff}}$) and effective electromechanical coupling coefficient ($K_{\mathrm{eff}}^{2}$) from the series and parallel resonance frequencies.

	Single Crystal Diamond				Polycrystalline Diamond			
Mode	$f_{s}\;\;(\mathrm{GHz})$	$f_{p}\;\;(\mathrm{GHz})$	$v_{eff}\;\;(m/s)$	$K_{eff}^{2}\;\;(\%)$	$f_{s}\;\;(\mathrm{GHz})$	$f_{p}\;\;(\mathrm{GHz})$	$v_{eff}\;\;(m/s)$	$K_{eff}^{2}\;\;(\%)$
Rayleigh	1.20	1.22	3425	3.46	1.20	1.22	3402	3.19
Sezawa	2.02	2.05	5725	3.72	2.06	2.09	5830	3.65
2nd Rayleigh	2.25	2.28	6361	3.04	2.27	2.30	6407	4.24

**Table 2 micromachines-13-01061-t002:** Quality factors and figure of merit (FOM) values from the series and parallel resonance frequencies.

	Single Crystal Diamond				Polycrystalline Diamond			
Mode	$Q_{s}$	$Q_{p}$	FOMs	FOMp	$Q_{s}$	$Q_{p}$	FOMs	FOMp
Rayleigh	251	187	8.69	6.49	103	4	3.28	0.14
Sezawa	52	69	1.91	2.58	10	10	0.376	0.372
2nd Rayleigh	67	132	2.03	4.02	62	8	2.63	0.336

**Table 3 micromachines-13-01061-t003:** Lower $\left(f_{1}\right)$ and higher $(f_{2}$) cut-off frequencies, center frequencies $\;\left(f_{c}\right)$, and bandwidth in the Rayleigh and Sezawa modes for SCD and PCD filters.

	Single Crystalline Diamond				Polycrystalline Diamond			
Mode	$f_{1}\;(\mathrm{GHz})$	$f_{2}\;(\mathrm{GHz})$	$\;f_{c}\;(\mathrm{GHz})$	−3 dB Bandwidth (MHz)	$f_{1}\;(\mathrm{GHz})$	$f_{2}\;(\mathrm{GHz})$	$\;f_{c}\;(\mathrm{GHz})$	−3 dB Bandwidth (MHz)
Rayleigh	2.14	2.21	2.17	72	2.61	2.72	2.66	107
Sezawa	4.81	4.99	4.90	181	4.64	4.83	4.74	189

**Table 5 micromachines-13-01061-t005:** Comparison of center frequencies, −3-dB bandwidth, insertion loss (IL), and the Q factor of the SAW filters with different substrates, piezoelectric thin films, and electrode metals.

Reference	Substrate	Piezoelectric Thin Film	Electrode Metal	Center Frequency (GHz)	−3 dB Bandwidth (MHz)	IL (dB)	Q
[35]	${\mathrm{SiO}}_{2}$/Si	AlN	Pt	4.47	30	–40	149
[36]	${\mathrm{SiO}}_{2}$/Si	${\mathrm{LiTaO}}_{3}$	Al	3.5	205	–1	17
[37]	PCD/Si	${\mathrm{SiO}}_{2}$/ZnO	Al	2.488	3	–5	700
[38]	PCD/Si	ZnO	Al	2.9	15	–20	193
This work	SCD/Si	${\mathrm{Sc}}_{0.43}{\mathrm{Al}}_{0.57}N$	Cr/Au	R 2.17–S 4.90	R 72–S 181	R − 2.5–S − 5	R 30–S 27
	PCD/Si	${\mathrm{Sc}}_{0.43}{\mathrm{Al}}_{0.57}N$	Cr/Au	R 2.66–S 4.74	R 107–S 189	R − 6–S – 6	R 25–S 25

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
