# Peer review of "SAW Resonators and Filters Based on Sc0.43Al0.57N on Single Crystal and Polycrystalline Diamond"

_micromachines, 2022, doi:10.3390/mi13071061_

Round 1

Reviewer 1 Report

The submitted manuscript is well written and the work about SAW resonators and filters based on Sc0.43Al0.57N is interesting. But there are several important things unclear. Suggestions and questions about this manuscript are as follows:

1-why the authors fabricate 1-port SAW resonators and SAW filters on ????? films with different thicknesses?

2-The single crystal diamond datasheet reports a surface roughness ?a value below 2 nm and the AFM analyses carried” Please add some Figures to show the results about materials characterization of ?????.

3-Line 35: It seems that the equations about veff and K2eff are incorrect, it should be veff=λ(fp+fs)/2, veff=π/2(fp-fs)/fp, please recalculate all the relevant results in table 1 and table 3 using the correct equations. Additionally, there is obvious format confusion in table 1. Please modified it carefully.

4-Line 197: what is the Q factor of SAW filter? How to calculate the Q factor of SAW filter? Besides, can the authors provide the Bode-Q of the fabricated resonators?

5- Figure 3. How many series and parallel resonators does the ladder type SAW filter consist? Is it on-wafer measurements or packaged and measured on EVB? Please provide a photo of the fabricated filter?

6-Again the information of the filters is significantly inadequate. Although the transmission characteristics of fabricated filters on both PCD/SCD are given in Figure 3, at least the detailed topology structure of the filters should be revealed for readers.   

7-In discussion section, the authors wrote “In layered structures, the effective propagation velocity of a SAW (3) depends on the elastic modulus…” What does the meaning of “SAW (3)” in the text?

Reviewer 2 Report

1. Need to add motivation and application of SAW device more.

2. Need to explain more about device preparation.

3. Need to add the measurement method more. I cannot clearly see how author get the data by measurement.

Reviewer 3 Report

The authors report fabrication and characterization of ScAlN-based SAW resonators and filters. The devices are well characterized and the discussion of prior work is satisfactory. I think this work merits publication in Micromachines provided the following points are addressed:

1. Please provide SEM characterization of the device that would complement Fig. 1

2. Please provide RBS data substantiating the claimed composition of the ScAlN thin film.

Minor comments:

1. Sc0.43Al0.57N should not be italicized. Variable are italicized but not chemical elements.

2. Please make sure all abbreviations are defined in the first instance of appearance. e.g. interdigital transducer (IDT), Rutherford backscattering spectrometry (RBS), etc.

Round 2

Reviewer 1 Report

Most of the previous review comments were well responded. But there still leaves some questions need be addressed

1the equations 1~2 is inappropriate. Here is some example as reference.

Veff

Huiling Liu et al, “Highly coupled leaky surface acoustic wave on hetero acoustic layer structures based on ScAlN thin films with a c-axis tilt angle”, 2021 Jpn. J. Appl. Phys. 60 031002

k2eff:

R. Lu, M. -H. Li, Y. Yang, T. Manzaneque and S. Gong, "Accurate Extraction of Large Electromechanical Coupling in Piezoelectric MEMS Resonators," in Journal of Microelectromechanical Systems, vol. 28, no. 2, pp. 209-218, April 2019, doi: 10.1109/JMEMS.2019.2892708.

2the equation for calculation of Bode Q-factor is not correct. Here is an example as reference.

Bode Q:

R. Ruby, "Review and comparison of bulk acoustic wave FBAR, SMR technology," in 2007 IEEE Ultrasonics Symposium Proceedings, 2007: IEEE, pp. 1029-1040.

3. A figure of merit (FOM) can be interesting to know since there is not a single design that has all highest properties such as coupling factor, shear phase velocity, quality factor, etc.

It says “A figure of merit usually employed to compare the performance of SAW devices is the Bode Q-factor. ”

However only using a Bode Q-factor may not be appropriate for FOM. FOM definition can be referred to a reference such as Zhang, et al., Progress in Materials Science, 68, 1-66 (2014).
